# SARS-CoV-2 seroprevalence and risk factors among meat packing, produce processing, and farm workers

**Melissa D. Klein**[1], **Michael Sciaudone**[2], **David Richardson**[3], **Roberto Lacayo**[2], **Colleen M. McClean**[4], **Oksana Kharabora**[2], **Katherine Murray**[2], **Miriana Moreno Zivanovich**[2], **Stephen Strohminger**[2], **Rachel Gurnett**[2], **Alena J. Markmann**[2], **D. Ryan Bhowmik**[5], **Emperatriz Morales Salgado**[2], **Edwin Castro-Arroyo**[6], **Allison E. Aiello**[3], **Ross M. Boyce**[3], **Jonathan J. Juliano**[2], **Natalie M. Bowman**[2]*

1 Department of Medicine, Washington University School of Medicine in St. Louis, St. Louis, Missouri, United States of America, 2 Department of Medicine, Division of Infectious Diseases, University of North Carolina at Chapel Hill School of Medicine, Chapel Hill, North Carolina, United States of America, 3 Department of Epidemiology, Gillings School of Global Public Health, University of North Carolina, Chapel Hill, North Carolina, United States of America, 4 School of Medicine, Duke University, Durham, North Carolina, United States of America, 5 Department of Microbiology and Immunology, University of North Carolina at Chapel Hill School of Medicine, Chapel Hill, North Carolina, United States of America, 6 University of North Carolina at Chapel Hill, Chapel Hill, North Carolina, United States of America

* natalie_bowman@med.unc.edu

**Data Availability Statement:** Original, de-identified data have been published for public access on the Harvard Dataverse (https://dataverse.harvard.edu/

## Abstract

Meat packing, produce processing, and farm workers are known to have an elevated risk of COVID-19, but occupational risk factors in this population are unclear. We performed an observational cohort study of meat packing, produce processing, and farm workers in North Carolina in fall 2020. Blood, saliva, and nasal turbinate samples were collected to assess for SARS-CoV-2 seropositivity. Risk factors for SARS-CoV-2 seropositivity were investigated using chi-square tests, two-sample t-tests, and adjusted risk ratio analyses. Among 118 enrolled workers, the baseline SARS-CoV-2 seroprevalence was 50.0%. Meat packing plant workers had the highest SARS-CoV-2 seroprevalence (64.6%), followed by farm workers (45.0%) and produce processing workers (10.0%), despite similar sociodemographic characteristics. Compared to SARS-CoV-2 seronegative workers, seropositive workers were more likely to work in loud environments that necessitated yelling to communicate (RR: 1.83, 95% CI: 1.25–2.69), work in cold environments (RR: 1.58, 95% CI: 1.12–2.24), or continue working despite developing symptoms at work (RR: 1.63, 95% CI: 1.14–2.32). After adjusting for age and working despite symptoms, high occupational noise levels were associated with a 1.72 times higher risk of SARS-CoV-2 seropositivity (95% CI: 1.16–2.55). Half of food processing workers showed evidence of past SARS-CoV-2 infection, a prevalence five times higher than most of the United States population at the time of the study. Work environments with loud ambient noise may pose elevated risks for SARS-CoV-2 transmission. Our findings also highlight the disproportionate burden of COVID-19 among underserved and economically disadvantaged Latinx communities in the United States.

dataset.xhtml?persistentId=doi:10.7910/DVN/ECIQXL&faces-redirect=true).

**Funding:** This study was supported by the North Carolina Collaboratory (NMB), the National Center for Advancing Translational Sciences (grant numbers 2KR27005 and 550KR242003, NMB), the National Institutes of Health (grant number UL1TR002489, NMB), and the National Institutes of Health SeroNet Serocenter of Excellence Award (grand number U54 CA260543, AJM). The funders had no role in study design, data collection and analysis, decision to publish, or preparation of the manuscript.

**Competing interests:** The authors have declared that no competing interests exist.

## Introduction

Early epidemiological studies of SARS-CoV-2, the causative agent of COVID-19, identified certain occupations that were associated with a high risk for infection [1]. Throughout the United States (US), meat packing plants, produce processing facilities, and commercial farms have served as foci for local outbreaks, especially in less densely populated rural areas [2–5]. Viral transmission in these facilities, sometimes resulting in hundreds of cases and necessitating plant closures, has been well publicized [6, 7]. As meat packing plants were declared "critical infrastructure" early in the COVID-19 pandemic, employees were expected to continue working despite the elevated risk [8]. Infected workers may subsequently infect their household members, contributing to spread in surrounding communities [9]. In addition to the adverse effects on human health, these localized outbreaks can overwhelm medical and public health institutions, especially in underserved rural communities where these industries are concentrated, and threaten the food supply due to worker absenteeism from illness [10].

Various hypotheses have been proposed for the elevated risk among meat packing, farming, and produce processing workers, referred to collectively as food processing workers. These include risk factors both inside and outside the workplace. Risk factors external to the workplace include social determinants of health such as living conditions, social behaviors, and shared transportation. These factors are strongly associated with the demographic characteristics of the food processing workforce, which is composed disproportionately of Black and Latinx individuals [2]. Factors specific to the workplace include close physical proximity between workers, prolonged shift durations, and shared spaces and equipment. Food processing facilities may also provide favorable physical environments for SARS-CoV-2 transmission due to cold temperatures, low humidity, metallic surfaces, and poor ventilation [11, 12]. Despite the reported incidence of SARS-CoV-2 among workers in food processing industries, our understanding of how to reduce viral transmission in this population is limited. The Centers for Disease Control and Prevention (CDC) and Occupational Safety and Health Administration (OSHA) have released interim guidance on reducing risk in the meat and poultry processing industry, but little data are available on the effectiveness or implementation of these guidelines [13]. In addition, it remains unclear whether the elevated risk of SARS-CoV-2 infection among food processing workers is due to socioeconomic factors or specific work conditions, or both.

Given the unique challenges and considerations for the prevention of COVID-19 transmission in the food processing industry, a targeted approach is warranted to prevent future outbreaks among workers and their families. Therefore, we conducted an observational cohort study of meat packing plant, farming, and produce processing workers in North Carolina, which has one of the largest food processing industries in the US [14]. Here we describe SARS-CoV-2 seroprevalence and identify characteristics associated with prior SARS-CoV-2 infection among workers, including occupational and socio-demographic risk factors.

## Methods

### Study population

Adults (age ≥18 years) who had resided and worked for at least two weeks in a meat packing plant, produce processing facility, or commercial farm in North Carolina since February of 2020 (when the first nontravel-related COVID-19 cases were identified in the US) were eligible to participate in the study. Exclusion criteria included age less than 18; inability to provide informed consent in English, Spanish, or a language for which an interpreter was available; and unwillingness to provide at least one blood sample.

## Data and specimen collection

Participants were recruited by contacting patients identified as potential food processing workers through retrospective chart review at Piedmont Health Services, a federally qualified health center (FQHC) with ten clinics in central North Carolina. In addition, we recruited through flyers, social media campaigns, and local community organizations. If eligible, participants were enrolled after providing informed consent in English or Spanish. Study visits took place at the study office or under a tent outside the participant's residence. Participants completed an enrollment visit between September and December 2020 consisting of a questionnaire to collect data on demographic characteristics, medical history, workplace and household characteristics, and preventive behaviors; three biological samples (blood, saliva, and nasal turbinate swab) were collected as well. Participants were screened for COVID-19 symptoms or exposures with weekly phone calls, and if necessary, referred to local clinics for free diagnostic testing. All study materials were available in English or Spanish and were administered by bilingual study personnel. Our study follows guidelines laid out in the STROBE statement [15].

## Sample processing

Whole blood samples were collected by venipuncture or finger prick in standard EDTA tubes (BD Cat # 367841). To separate plasma, the whole blood samples were centrifuged within 24 hours of collection at 1600g for 30 minutes. Prior to serological testing, all plasma specimens were heat-inactivated at 56˚C for 30 minutes to reduce risk from any residual virus, mixing the sample by inverting the tube every 5 minutes. The inactivated samples were centrifuged at 1500g for 10 minutes, and the supernatant was aliquoted and stored at -80˚C until further testing.

## Serological studies

We assessed past infection with SARS-CoV-2 using an established total Ig SARS-CoV-2 RBD (receptor binding domain) enzyme-linked immunosorbent assay (ELISA) [16]. The spike protein N-terminal domain (NTD) antigen (16–305 amino acids, Accession: P0DTC2.1) was cloned into the pαH mammalian expression vector and purified using nickel-nitrilotriacetic acid agarose in the same manner. To briefly summarize the ELISA, 50 μl of spike RBD antigen at 4 μg/ml in Tris Buffered Saline (TBS) pH 7.4 was coated in the 96-well high-binding microtiter plate (Greiner Bio-One cat # 655061) for 1 hour at 37˚C. Then the plate was washed three times with 200 μl of wash buffer (TBS containing 0.2% Tween 20) and blocked with 100 μl of blocking solution (3% milk in TBS containing 0.05% Tween 20) for 1 hour at 37˚C. The blocking solution was removed, and 50 μl of serum sample at 1:20 or indicated dilutions in blocking buffer was added for 1 hour at 37˚C. The plate was washed in the wash buffer, 50 μl of alkaline phosphatase-conjugated secondary goat anti-human secondary antibody at 1:2500 dilution was added for 1 hour at 37˚C. For measuring total Ig, a mixture of anti-IgG (Sigma Cat # A9544), anti-IgA (Abcam Cat # AB97212), and anti-IgM (Sigma Cat # A3437) were added together. The plate was washed, and 50 μl p-Nitrophenyl phosphate substrate (SIGMA FAST, Cat No N2770) was added to the plate and absorbance measured at 405nm using a plate reader (Biotek Epoh, Model # 3296573). Optical density (OD) was measured with a VICTOR Nivo multimode plate reader (PerkinElmer, Waltham, Massachusetts) at 3, 5, 7, 9, and 11 minutes after substrate was added. Samples were tested in duplicate, and duplicate values with variance >25% and/or one value above assay cutoff were repeated.

Seventeen participants were unable to provide blood by venipuncture, so capillary blood was collected using a finger prick method yielding <1 mL blood. These samples could not be

separated by centrifugation and subsequently hemolyzed during storage. Since our plasma-based ELISA was inappropriate for these samples, antibody was detected using UNscience COVID-19 IgG/IgM rapid detection test strips; results were confirmed using a technique based on a previously described whole blood ELISA [17]. The rapid detection test results were confirmed and quantified using the same protocol parameters from the previous plasma-based ELISA through the substitution of whole blood for plasma and the addition of reconstituted whole blood controls for quality control. The reconstituted whole blood controls involved 55% plasma control and 45% erythrocytes to mimic blood proportions, which were further diluted into blocking buffer until the plasma control matched the sample dilutions on the plate. Whole blood ELISA titers of 1:20, 1:40, and 1:120 were compared to plasma-based ELISA OD readings using confirmed SARS-CoV-2 antibody negative and positive controls. The sensitivity and specificity of the 1:20 titer were previously obtained [18]. We calculated the sensitivity and specificity of the 1:40 and 1:120 titers (S1 Table).

## Statistical analysis

The primary outcome of interest was SARS-CoV-2 seropositivity. Potential explanatory variables of interest are shown in Table 2. We classified each worker as employed in meat packing, farming, or produce processing. Two-sample Student's t-tests and chi-square tests were used to evaluate characteristics divided by industry (Table 1). In this analysis, we only directly compared the meat packing and farming industries due to sparse data in the produce processing industry. Next, we calculated unadjusted risk ratios using a log-binomial model and p-values using chi-square tests (when risk ratios were inappropriate) to examine the association between occupational risk factors and SARS-CoV-2 seropositivity (Table 2). We also calculated adjusted risk ratios using a traditional stepwise regression approach [19]. Variables significantly associated with the outcome in the unadjusted analysis were included as covariates; we also included age due to its significance in the existing literature [20]. We calculated adjusted risk ratios with and without highly correlated variables (Model 2 and 1, respectively). Finally, we evaluated worker characteristics divided by industry, using descriptive statistics, risk ratios, and chi-square tests (S1 Table). Again, produce processing workers were excluded due to sparse data. A p-value below 0.05 was considered statistically significant. Data were analyzed using Stata version 16 (StataCorp, College Park, Maryland).

## Ethics

This study was approved by the Institutional Review Board at the University of North Carolina at Chapel Hill (IRB 20–2032). Written informed consent was obtained from participants in English or Spanish by bilingual study personnel. Participant confidentiality was maintained by removing personal identifiers from datasets and biospecimens. Participant data with personal identifiers were stored in a locked cabinet only accessible to authorized study personnel. Electronic participant data did not include identifiers and were only accessible through a password-protected REDCap database to authorized study personnel.

## Results

From September to December 2020, we enrolled 118 participants who provided blood samples. Participants were 61.0% female and 94.9% Latinx, with a mean age of 41.9 (± 13.2) years. Sixty (50.8%) worked on farms, 48 (40.7%) worked at meat packing facilities, and 10 (8.5%) worked in produce processing. Demographics including age at enrollment, sex, and current smoking status were similar between industries (Table 1). Latinx individuals represented a

**Table 1. North Carolina food processing worker characteristics by industry (fall 2020).**

| | Meat packing (n = 48) | Farming (n = 60) | Produce processing (n = 10) | P-value (meat packing vs. farming) |
|---|---|---|---|---|
| | Mean ± SD or n (%) | | | |
| Age | 44 ± 12 | 40 ± 12 | 40 ± 20 | 0.06 |
| Female sex | 29 (60%) | 35 (58%) | 8 (80%) | 0.83 |
| Latinx ethnicity | 42 (88%) | 60 (100%) | 10 (100%) | **0.005** |
| Current smoker | 4 (8%) | 4 (7%) | 0 (0%) | 0.74 |
| Frequently socialize in person | 5 (10%) | 3 (5%) | 1 (10%) | 0.29 |
| Full time status | 41 (85%) | 41 (68%) | 5 (50%) | **0.04** |
| Weeks employed | 30 ± 11 | 27 ± 15 | 22 ± 11 | 0.21 |
| Hours per week | 43 ± 5 | 41 ± 13 | 38 ± 9 | **0.01** |
| Carpool to work | 17 (35%) | 24 (40%) | 3 (30%) | 0.83 |
| Mask wearing | 43 (90%) | 40 (67%) | 4 (40%) | **0.005** |
| Eye protection | 20 (42%) | 9 (15%) | 4 (40%) | **0.002** |
| Protective clothing | 42 (88%) | 17 (28%) | 30 (30%) | **<0.001** |
| Shoe covers | 11 (23%) | 2 (3%) | 30 (30%) | **0.002** |
| Shield | 31 (65%) | 8 (13%) | 10 (100%) | **<0.001** |
| Clean personal items | 19 (40%) | 26 (43%) | 30 (30%) | 0.69 |
| Temperature check | 38 (79%) | 18 (30%) | 3 (30%) | **<0.001** |
| Symptom check | 5 (10%) | 2 (5%) | 2 (20%) | 0.29 |
| Work despite developing symptoms | 10 (21%) | 6 (10%) | 0 (0%) | 0.12 |
| Indoors | 42 (88%) | 19 (32%) | 6 (60%) | **<0.001** |
| Crowded | 30 (62%) | 9 (15%) | 40 (20%) | **<0.001** |
| Cold temperature | 37 (77%) | 2 (3%) | 2 (20%) | **<0.001** |
| High noise level | 36 (75%) | 16 (27%) | 4 (40%) | **<0.001** |
| SARS-CoV-2 seropositive | 31 (65%) | 27 (45%) | 1 (10%) | **0.04** |

In this analysis, we only directly compared the meat packing and farming industries due to sparse data in the produce processing industry.

Frequently socialize in person: socializing with friends in person (outside of school or work) at least several days per week. Weeks employed: number of weeks worked at current place of employment in 2020. Hours: average number of hours worked per week at current place of employment. Carpool to work: gets to work by driving with at least one other person or by using a ride-share van or shuttle vehicle. Mask wearing: regularly wearing a mask at work. Eye protection: regularly wearing protective glasses or face shields at work. Protective clothing: regularly wearing protective clothing (such as a gown or apron) at work. Shoe covers: regularly wearing shoe covers at work. Shield: having a protective shield between the individual and other workers. Frequent hand washing: washing or sanitizing hands multiple times per day at work. Clean personal items: cleaning personal items (such as phone or keys) at work. Temperature check: always required to complete a temperature check before each shift. Symptom check: always required to complete a symptom check before each shift. Indoors: always working indoors. Cold temperatures: below 60°F or 16°C. Crowded: being close enough to touch another worker without walking. High noise level: loud enough to necessitate yelling to communicate.

modestly higher proportion of farm workers than meat packing workers (100% vs 88%, p 0.005).

The overall prevalence of SARS-CoV-2 antibodies in our study population was 50.0%. SARS-CoV-2 seroprevalence was highest among meat packing plant workers (64.6%), followed by farm workers (45.0%) and produce processing workers (10.0%). Age, sex, ethnicity, smoking status, frequency of in-person socializing, full time employment status, hours worked per week, and duration of employment were similar between seropositive and seronegative workers (Table 2).

Temperature and symptom checks before shifts were always required for 50.4% and 8.5% of workers, respectively. Consistent temperature checks were most common among meat packing plant workers (79.2%) and least common among farm workers (30.0%). No participants reported working after failing a temperature or symptom check; however, 16 (13.6%) workers reported developing symptoms such as fever, cough, or shortness of breath while

**Table 2. North Carolina food processing worker and workplace characteristics associated with SARS-CoV-2 seropositivity (fall 2020).**

| | Seropositive (n = 59) | Seronegative (n = 59) | RR (95% CI) | aRR (95% CI): Model 1* | aRR (95% CI): Model 2† |
|---|---|---|---|---|---|
| | Mean ± SD or n (%) | | | | |
| Age | 40 ± 12 | 44 ± 15 | 0.99 (0.98–1.00) | 1.00 (0.98–1.01) | 0.99 (0.98–1.01) |
| Female sex | 35 (59) | 37 (63) | 0.93 (0.65–1.34) | | |
| Latinx ethnicity | 57 (97) | 55 (93) | 1.53 (0.49–4.80) | | |
| Current smoker | 2 (3) | 6 (10) | 0.48 (0.14–1.62) | | |
| Frequently socialize in person | 4 (7) | 5 (8) | 0.87 (0.41–1.85) | | |
| Full-time employee | 45 (76) | 42 (71) | 1.15 (0.74–1.77) | | |
| Mean weeks employed | 28 ± 14 | 27 ± 14 | 1.00 (0.99–1.01) | | |
| Mean hours per week | 42 ± 6 | 41 ± 12 | 1.00 (0.99–1.02) | | |
| Carpool to work | 23 (39) | 21 (36) | 0.76 (0.54–1.07) | | |
| Temperature checks | 29 (49) | 30 (51) | 0.95 (0.66–1.36) | | |
| Symptom checks | 6 (10) | 4 (7) | 1.21 (0.71–2.08) | | |
| Working despite developing symptoms | 12 (20) | 4 (7) | **1.63 (1.14–2.32)** | 1.27 (0.87–1.85) | 1.28 (0.92–1.78) |
| Indoors | 38 (64) | 29 (49) | 1.38 (0.93–2.03) | | |
| Crowded environment | 25 (42) | 18 (31) | 1.28 (0.90–1.83) | | |
| Cold environment | 27 (46) | 14 (24) | **1.58 (1.12–2.24)** | | 1.27 (0.86–1.87) |
| High noise level | 37 (63) | 19 (32) | **1.83 (1.25–2.69)** | **1.72 (1.16–2.55)** | 1.49 (0.94–2.39) |
| Meat packing | 31 (53) | 17 (29) | **1.61 (1.13–2.30)** | - | - |
| Farming | 27 (46) | 33 (56) | 0.82 (0.57–1.17) | - | - |
| Produce processing | 1 (2) | 9 (15) | 0.19 (0.29–1.21) | - | - |

*Model 1: Included age, working despite developing symptoms, and high noise level

†Model 2: Included age, working despite developing symptoms, cold environment, and high noise level

aRR: adjusted risk ratio; CI: confidence interval; RR: risk ratio; SD: standard deviation. Frequently socialize in person: socializing with friends in person (outside of school or work) at least several days per week. Weeks employed: number of weeks worked at current place of employment in 2020. Hours per week: number of hours worked per week at current place of employment. Carpool to work: gets to work by driving with at least one other person or by using a ride-share van or shuttle vehicle. Mask wearing: regularly wearing a mask at work. Eye protection: regularly wearing protective glasses or face shields at work. Protective clothing: regularly wearing protective clothing (such as a gown or apron) at work. Shoe covers: regularly wearing shoe covers at work. Shield: having a protective shield between the individual and other workers. Frequent hand washing: washing or sanitizing hands multiple times per day at work. Clean personal items: cleaning personal items (such as phone or keys) at work. Temperature check: always required to complete a temperature check before each shift. Symptom check: always required to complete a symptom check before each shift. Std dev: standard deviation.

working but continuing to work. Continuing to work after developing symptoms was associated with SARS-CoV-2 seropositivity (RR: 1.63, 95% CI: 1.14–2.32) and was most common among meat packing workers (Tables 1 and 2).

Regular mask wearing and eye protection at work were reported by 77.5% and 32.5% of participants, respectively, and 75.8% of workers reported washing their hands at work several times per day. Over a third (36.4%) of workers reported working in crowded conditions close enough to touch another worker without walking. Meat packing workers more likely than other industry workers to wear masks at work, wear protective clothing, use shields between workers, or have required temperature checks before shifts.

Prior COVID-19 illness, defined as a history of typical symptoms such as fever and cough in 2020, was reported by 35 participants (29.7%). Of the 59 participants with positive SARS-CoV-2 serology, 29 (49.2%) reported a history of COVID-19, compared to 6 (10.2%) of the 59 seronegative participants. Among seropositive workers, farm workers were significantly less likely than other industry workers to report a history of COVID-19 (22.2% vs. 71.9%, p<0.001).

Indoor, crowded, cold, and loud work environments were highly correlated with each other. High noise levels (defined as being loud enough to necessitate yelling to communicate) and cold work environments (defined as ambient temperature below 16˚C) were significantly associated with SARS-CoV-2 seropositivity (RR: 1.83, 95% CI: 1.25–2.69 and RR: 1.58, 95% CI: 1.12–2.24, respectively) (Table 2). Adjusted for age and working despite symptoms, high noise levels were associated with 1.72 times the risk of SARS-CoV-2 seropositivity (95% CI: 1.16–2.55) (Table 2). Due to its high correlation with high noise levels, cold work environment was not included in the first adjusted model (Model 1). Notably, the association between loud environments and SARS-CoV-2 seropositivity remained significant in the subset of workers who did not work in cold environments (RR: 1.68, 95% CI: 1.02–2.77). Similar trends were observed in workers stratified by industry (S2 Table), although the association between high noise levels and SARS-CoV-2 seropositivity only met statistical significance among farm workers (RR: 1.23, 95% CI: 1.02–1.49). Shoe covers were associated with higher SARS-CoV-2 seropositivity among meat packing workers alone, and shields between workers were associated with higher SARS-CoV-2 seropositivity among farm workers alone (S2 Table).

Because working in cold environments was almost perfectly correlated with working inside, we also examined the effect of working indoors among those who did not work in a cold environment. Among workers who did not report working in cold environments, working indoors was not associated with SARS-CoV-2 seroprevalence (p = 0.79).

## Discussion

The prevalence of SARS-CoV-2 antibodies in our study population was 50.0%, over five times the rate observed in most of the United States when we conducted this study in fall 2020 [21]. Meat processing workers had the highest prevalence, with almost two-thirds showing evidence of past SARS-CoV-2 infection. Approximately half of participants with positive SARS-CoV-2 serology reported a known history of COVID-19, suggesting a high rate of asymptomatic infections, corroborating other studies [22]. Farm workers were most likely to have SARS-CoV-2 antibodies without a reported history of COVID-19 illness, possibly reflecting lower access to testing. This is consistent with mass testing efforts at certain meat packing plants with known outbreaks in spring 2020, which may have resulted in higher testing rates in this population [23]. Farm workers also frequently experience non-specific respiratory symptoms and fatigue from occupational exposures that could be difficult to distinguish from COVID-19 illness [24].

Racial and ethnic minorities have suffered a disproportionate burden of COVID-19 morbidity and mortality, and similar trends have been seen among food processing workers [25, 26]. In the early months of the pandemic, 87% of known COVID-19 cases among agricultural workers occurred in racial or ethnic minorities, particularly Latinx communities [2]. Many of these workers are recent, sometimes undocumented, immigrants to the United States, lack political and social capital to advocate for worker safety, and have limited access to health care. Food processing workers are particularly vulnerable as they may face stressful and physically dangerous work, low wages, and limited benefits [27, 28]. Our results further demonstrate the disproportionate burden of COVID-19 among underserved and economically disadvantaged Latinx communities in the U.S.

To reduce SARS-CoV-2 transmission in food processing industries, the CDC and OSHA have recommended a number of policies such as mandatory mask wearing, social distancing between workers, frequent hand hygiene, and adequate sick leave policies to ensure workers are not penalized for reporting illness [29]. However, little if any enforcement exists for these guidelines. Our findings raise concern that inadequate safety guidelines in many food

processing workplaces may have contributed to the high SARS-CoV-2 seroprevalence among workers, although many workers may have been infected early in the pandemic before these guidelines were enacted. Notably, 13.6% of workers reported developing symptoms concerning for COVID-19 during a shift but continued working. This finding highlights the importance of adequate sick leave policies for workers and policies to protect workers who report illness.

High noise levels appeared to be the primary occupational risk factor for SARS-CoV-2 seropositivity among workers. Workers who always worked in loud environments were at a 1·7 times higher risk of SARS-CoV-2 seropositivity. This association remained significant when limited to workers who did not work in cold environments, further supporting the importance of high noise levels as a risk factor. In addition, among farming and produce processing workers who reported always working in loud areas, we observed a SARS-CoV-2 seroprevalence comparable to that observed in the meat packing industry. To our knowledge, this is the first study demonstrating an association between loud work environments and SARS-CoV-2 infection risk. High noise levels may require workers to reduce distancing, shout, or remove face masks to communicate, increasing respiratory droplet emission. This association may not be due directly to high noise levels, as volume may be a surrogate marker of jobs that are high risk for other reasons. A cold work environment was also associated with higher risk of SARS-CoV-2 seropositivity in unadjusted analysis. A few studies have identified increased stability of the SARS-CoV-2 virus at cold temperatures, which could enhance aerosol or fomite-mediated transmission [30]. However, high noise level was associated with a greater magnitude of increased risk than cold environment in both unadjusted and adjusted analyses and is likely a more easily modifiable risk factor in the workplace. Our results suggest that mitigation strategies, such as reducing ambient noise levels, use of speakers, optimized facility engineering, and consistent mask use, could help reduce transmission risk in both food processing facilities and other high-risk work environments [31].

Our study identified an association between SARS-CoV-2 seropositivity and shoe covers among meat packing workers. One possible explanation for this association is that a lack of proper training on donning and doffing methods of personal protective equipment (PPE) may inadvertently lead to self-contamination of clothes and skin. However, it is more likely that particular meat packing jobs requiring shoe covers imparted a higher risk of SARS-CoV-2 transmission due to additional factors we were unable to measure, such as prolonged close contact between workers or less frequently sanitized work areas. Some meat packing facilities may also have implemented additional safety measures such as shoe covers after outbreaks. This is supported by the fact that the association between shoe covers and SARS-CoV-2 seropositivity was not observed among farming and food processing workers. Similarly, we observed an association between shield use and SARS-CoV-2 seropositivity among farm workers alone, which is likely also explained by unmeasured risk factors of jobs involving shields between workers.

Our study did not observe a significant association between reported mask wearing at work and SARS-CoV-2 seropositivity, but our surveys did not capture the technique with which masks were worn or mask type. Given the substantial evidence in existing literature that masks reduce the spread of SARS-CoV-2, our findings raise concern that many food processing workers may have poor quality masks or wear them with improper technique, such as below the nose [32]. Additionally, it is likely that many participants were infected several months before the study started, as there were known outbreaks at several plants in the spring of 2020, when mask hygiene may not have been as widespread in the workplace or community.

Our study has several limitations. We used a convenience sample that may not accurately reflect the true seroprevalence of past SARS-CoV-2 infections in this population. In addition,

our study could not include individuals who died of COVID-19, though this is a small proportion of affected food processing workers [2, 33]. Behavioral data was collected by self-report, which is susceptible to a variety of biases such as recall and social desirability bias. Some workers may have been unwilling to participate due to concerns about their results affecting job security, although we assured participants their surveys and test results would remain confidential and did not recruit participants directly through employers to reduce this risk. It is also possible that our ELISA missed a few early cases due to antibody waning. We estimate this represents a very small number of our participants, as all participants were tested within 10 months of the first identified COVID-19 case in North Carolina, and prior studies have demonstrated durable antibody response for at least six to nine months [34, 35]. Furthermore, current behavior may not reflect accurately behaviors performed earlier in the COVID-19 pandemic, as public health guidelines evolved rapidly. Finally, our study population was limited to rural, almost entirely Latinx workers that may not be representative of food processing and farm worker populations in other states or countries.

In summary, our study identified a remarkably high seroprevalence of SARS-CoV-2 among meat packing, produce processing, and farm workers and their household members. These findings highlight the burden of the COVID-19 pandemic in this vulnerable population and the role of food processing facilities in community spread. This is the first study to examine occupational risk factors for SARS-CoV-2 seropositivity among meat packing, produce processing, and farm workers. Occupational noise appeared strongly associated with SARS-CoV-2 seropositivity, which has notable implications for policy recommendations and facility engineering. Although COVID-19 vaccines have become widely accessible in the US, understanding occupational risk factors remains important given incomplete vaccination rates, the emergence of new variants, and the likelihood of future airborne epidemics. This study also highlights the impacts of occupational disadvantage in the setting of COVID-19, experienced predominantly by ethnic minorities in a high-risk industry. Further research is warranted to investigate the effectiveness of mitigation strategies to reduce SARS-CoV-2 transmission among food processing workers.

## Supporting information

**S1 Table. ELISA validation data.**
(DOCX)

**S2 Table. Food processing occupational characteristics associated with SARS-CoV-2 seropositivity stratified by industry (fall 2020).**
(DOCX)

## Acknowledgments

We wish to thank the staff of Piedmont Health Services, the Hispanic Liaison, and the Episcopal Farmworker Ministry, who collaborated in the study. Premkumar Lakshmanane provided valuable advice on serological assays. We also thank Maggie Oberly and Suvra Mitra for their assistance with laboratory work. Most importantly, we thank the workers and their household members for their valuable participation and time in this study.

## Author Contributions

**Conceptualization:** Melissa D. Klein, Michael Sciaudone, David Richardson, Allison E. Aiello, Ross M. Boyce, Jonathan J. Juliano, Natalie M. Bowman.

**Data curation:** Melissa D. Klein, Michael Sciaudone.

**Formal analysis:** Melissa D. Klein, David Richardson, Natalie M. Bowman.

**Funding acquisition:** Melissa D. Klein, Michael Sciaudone, Natalie M. Bowman.

**Investigation:** Melissa D. Klein, Michael Sciaudone, Roberto Lacayo, Colleen M. McClean, Miriana Moreno Zivanovich, Stephen Strohminger, Rachel Gurnett, Alena J. Markmann, Emperatriz Morales Salgado, Edwin Castro-Arroyo, Natalie M. Bowman.

**Methodology:** David Richardson, Oksana Kharabora, Katherine Murray, D. Ryan Bhowmik, Allison E. Aiello, Natalie M. Bowman.

**Project administration:** Melissa D. Klein, Michael Sciaudone, Natalie M. Bowman.

**Resources:** Oksana Kharabora, Alena J. Markmann, Ross M. Boyce, Jonathan J. Juliano, Natalie M. Bowman.

**Supervision:** Melissa D. Klein, Natalie M. Bowman.

**Validation:** Oksana Kharabora.

**Writing – original draft:** Melissa D. Klein, Michael Sciaudone, David Richardson, Natalie M. Bowman.

**Writing – review & editing:** Melissa D. Klein, Michael Sciaudone, David Richardson, Roberto Lacayo, Colleen M. McClean, Oksana Kharabora, Katherine Murray, Miriana Moreno Zivanovich, Stephen Strohminger, Rachel Gurnett, Alena J. Markmann, D. Ryan Bhowmik, Emperatriz Morales Salgado, Edwin Castro-Arroyo, Allison E. Aiello, Ross M. Boyce, Jonathan J. Juliano, Natalie M. Bowman.

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
