## [Decision Letter · Decision Letter 0]

9 Feb 2022

PGPH-D-21-01092

SARS-CoV-2 Seroprevalence and Risk Factors Among Meat Packing, Produce Processing, and Farm Workers

Dear Dr. Klein,

Thank you for submitting your manuscript to PLOS Global Public Health. After careful consideration, we feel that it has merit but does not fully meet PLOS Global Public Health’s publication criteria as it currently stands. Therefore, we invite you to submit a revised version of the manuscript that addresses, point by point, the comments of the reviewers.

We look forward to receiving your revised manuscript.

Kind regards,

Justin V. Remais

Academic Editor

Journal Requirements:

1. We have noticed that you have cited a Supplement in the manuscript file but there is corresponding file uploaded. Please amend this and add Legends after the References Section.

2. In the online submission form, you indicated that "Original data are available upon request.". All PLOS journals now require all data underlying the findings described in their manuscript to be freely available to other researchers, either 1. In a public repository, 2. Within the manuscript itself, or 3. Uploaded as supplementary information.

3. Please amend your detailed Financial Disclosure statement. This is published with the article, therefore should be completed in full sentences and contain the exact wording you wish to be published.

ii). State the initials, alongside each funding source, of each author to receive each grant.

iii). State what role the funders took in the study. If the funders had no role in your study, please state: “The funders had no role in study design, data collection and analysis, decision to publish, or preparation of the manuscript.”

Additional Editor Comments (if provided):

Reviewers' comments:

Reviewer's Responses to Questions

**Comments to the Author**

1. Does this manuscript meet PLOS Global Public Health’s publication criteria? Is the manuscript technically sound, and do the data support the conclusions? The manuscript must describe methodologically and ethically rigorous research with conclusions that are appropriately drawn based on the data presented.

Reviewer #1: Yes

Reviewer #2: Yes

2. Has the statistical analysis been performed appropriately and rigorously?

Reviewer #1: Yes

Reviewer #2: Yes

3. Have the authors made all data underlying the findings in their manuscript fully available (please refer to the Data Availability Statement at the start of the manuscript PDF file)?

Reviewer #1: Yes

Reviewer #2: Yes

4. Is the manuscript presented in an intelligible fashion and written in standard English?

Reviewer #1: Yes

Reviewer #2: Yes

5. Review Comments to the Author

Reviewer #1: I wish to commend the authors on their well written manuscript that was a delight to read. The authors have completed a noteworthy analysis of the risk factors of SARS-CoV-2 infection among a key occupational group in an insightful and considerate way. Small mentions of this appear in the methods that demonstrate the lengths the authorship team went to gather data. First starting with enrollment at a federally qualified health center, then engaging local community organizations and traveling to participants residences to for study visits. The authors also followed their participants weekly and referred them to local clinics for additional testing. This study demonstrates a careful and considerate analysis of a vulnerable population while also significantly contributing to the literature.

I offer a few suggestions that I think will make the manuscript stronger.

1. Major concern: The authors mention repeat biological sampling from their study participants and it is not clear what samples were used in the analysis or the date/range for the estimates of seroprevalence. For example, did the authors consider all samples when evaluating seroprevalence of SARS-CoV-2 or just the first one w/ enrollment. It is not clear in the manuscript as written, what time window their results are from and as a result, the conclusions will have limited impact in this extended pandemic with multiple variants and shifting measurement/containment practices if it is not clear what the exact time period or range the authors have measured.

a. Example: Statistical analysis: “The primary outcome of interest was SARS-CoV-2 positivity” by what time point?

b. The authors mention in the first time of discussion “when we conducted this study in September 2020- but in another section of the paper refer to September – December for enrollment of participants.

c. It appears as if the data analyzed do not include the monthly follow-up visits. Should these results be included in the methods if they are not used?

d. Enrollment criteria – working since “February 2020” I imagine this is just the cut off for being employed prior to the start of the pandemic but it might be good to explicitly state this.

2. I think the authors have placed a bit too much emphasis on statistical significance given their small sample size. “When cold work environment was added to the adjusted analysis (Model 2), the association of SARS-CoV-2 seropositivity with high noise levels and cold environment lost statistical significance.” While this is true the magnitude was still notable and the lower CI was 0.94. As written, with no additional interpretation- it dismisses the association as if it were null. It is good to caution against over interpreting- but the authors see attenuation of their association which is expected.

3. Tables need more descriptive titles such that they can stand alone from the paper. Please consider adding dates, locations and other descriptive measures of the study population to the table. Also please see comment 1 about when exactly seroprevalence was measured- this should be clear across the manuscript text and in the table(s) and supplement

4. Authors mention a supplement but there was no supplement file available in the peer review system. As such- I was unable to review the supplemental results.

Minor thoughts

1. When farm workers were most likely to have SARS-CoV-2 antibodies without a history of COVID-19 illness- is there any literature or anecdotal knowledge about the inability to distinguish low/moderate symptoms of COVID-19 illness with other farm laborer sickness (i.e. pesticide exposure, heat stress) or work-related fatigue? I wonder if that may also be a reason why fewer illnesses were reported in this group

2. Also relevant to discussion- are there any studies/anecdotal knowledge about cold environments and nasal discharge (runny noses) that also may increase the spread of COVID-19?

3. Page 4- liner 101-102 “Which has one of the food processing industries in the US” – seems like a word or two is missing from this phrase. Please revise for clarity.

Reviewer #2: This manuscript furthers understanding of COVID-19 transmission in food processors, a large yet understudied group. Of particular interest and importance is the linkage between noise levels and transmission. See the attached documents for comments and minor revisions.

6. PLOS authors have the option to publish the peer review history of their article (what does this mean?). If published, this will include your full peer review and any attached files.

**Do you want your identity to be public for this peer review?** For information about this choice, including consent withdrawal, please see our Privacy Policy.

Reviewer #1: No

Reviewer #2: No

---

## [Editor Report · Decision Letter 1]

27 May 2022

SARS-CoV-2 Seroprevalence and Risk Factors Among Meat Packing, Produce Processing, and Farm Workers

PGPH-D-21-01092R1

Dear Dr. Klein,

We are pleased to inform you that your manuscript 'SARS-CoV-2 Seroprevalence and Risk Factors Among Meat Packing, Produce Processing, and Farm Workers' has been provisionally accepted for publication in PLOS Global Public Health.

Best regards,

Miguel Angel Garcia-Bereguiain, PhD

Academic Editor